# The Sex Difference in 6-h Ultra-Marathon Running—The Worldwide Trends from 1982 to 2020

**DOI:** 10.3390/medicina58020179

**Published:** 2022-01-25

**Authors:** Beat Knechtle, Katja Weiss, Elias Villiger, Volker Scheer, Thayse Natacha Gomes, Robert Gajda, Nejmeddine Ouerghi, Hamdi Chtourou, Pantelis T. Nikolaidis, Thomas Rosemann, Mabliny Thuany

**Affiliations:** 1Medbase St. Gallen Am Vadianplatz, 9000 St. Gallen, Switzerland; katja@weiss.co.com; 2Institute of Primary Care, University of Zurich, 8000 Zurich, Switzerland; thomas.rosemann@usz.ch; 3Klinik für Allgemeine Innere Medizin, Kantonsspital St. Gallen, 9000 St. Gallen, Switzerland; evilliger@gmail.com; 4Ultra Sports Science Foundation, 69310 Pierre-Benite, France; volkerscheer@yahoo.com; 5Department Sports and Health, Institute of Sports Medicine, Paderborn University, 33098 Paderborn, Germany; 6Department of Physical Education, Federal University of Sergipe, São Cristóvão 49100-000, Brazil; thayse_natacha@hotmail.com; 7Center for Sports Cardiology, Gajda-Med Medical Center in Pułtusk, 06-100 Pułtusk, Poland; gajda@gajdamed.pl; 8Department of Kinesiology and Health Prevention, Jan Dlugosz University in Częstochowa, 42-200 Częstochowa, Poland; 9High Institute of Sport and Physical Education of Kef, University of Jendouba, UR13JS01, Kef 7100, Tunisia; najm_ouerghi@hotmail.com; 10Faculty of Medicine of Tunis, Rabta Hospital, University of Tunis El Manar, LR99ES11, Tunis 1007, Tunisia; 11Institut Supérieur du Sport et de l’Education Physique de Sfax, Université de Sfax, Sfax 3000, Tunisia; h_chtourou@yahoo.fr; 12Activité Physique, Sport et Santé, UR18JS01, Observatoire National du Sport, Tunis 1003, Tunisia; 13School of Health and Caring Sciences, University of West Attica, 12243 Athens, Greece; pademil@hotmail.com; 14Centre of Research, Education, Innovation and Intervention in Sport (CIFI2D), Faculty of Sport, University of Porto, 4200-450 Porto, Portugal; mablinysantos@gmail.com

**Keywords:** endurance, ultra-running, age of peak performance, nationality

## Abstract

*Background and Objectives:* The 6-h ultra-marathon is the shortest time-limited ultra-marathon race, but little has been investigated regarding this race format. Previously, only the age of peak performance in the context of longer time-limited ultra-marathons was determined. The purpose of this study was to investigate the trends in 6-h ultra-marathon races from 1982 to 2020 for female and male ultra-runners, the participation and performance by countries, the age of peak performance, and the differences in performance regarding countries. *Materials and Methods:* The sample included 23,203 female ultra-runners, aged 18–83 years, and 87,264 male ultra-runners, aged 18–85 years, who were finishers in a 6-h ultra-marathon held between 1982 and 2020. The age of peak performance was tested using the Kruskal–Wallis test, followed by the Bonferroni Correction. The difference in performance by countries was verified using a linear regression model with the fastest runners from Russia in women, and Tunisia in men, used as reference. *Results:* Over the years, the men-to-women ratio decreased. The mean age was 43.20 ± 9.30 years for female and 46.09 ± 10.17 years for male runners. Athletes in younger age groups were faster than athletes in older age groups. Most female and male participants originated from Germany. Women from Russia (10.01 ± 1.28 km/h) and men from Tunisia (12.16 ± 1.46 km/h) were the fastest. *Conclusions:* In summary, in 6-h ultra-marathons held between 1982 and 2020, the participation for both women and men increased, while the men-to-women ratio decreased. The mean age was higher in men compared to women. Most female and male runners originated from Germany, but the fastest women were from Russia, and the fastest men from Tunisia. Future studies need to investigate whether Russian women and Tunisian men are also the best in other distance-limited ultra-marathon races, such as 12-h and 24-h.

## 1. Introduction

Ultramarathon races are distance- (e.g., 180 km, 360 km) or time-limited (e.g., 6 h to 10 days) events [1], and have dramatically increased in popularity in recent years [2]. Aiming to better understand and describe the profile of these athletes, several studies investigated a plethora of variables, such as possible predictors of the performance [3,4,5], the age of peak performance [6], the athletes’ sociodemographic characteristics [7,8], and the trend of participation [9]. 

Regarding the participation trends in ultra-marathon running, Cejka et al. showed an increase in the number of finishers from Japan, Germany, Poland, USA, and Italy between 1998 and 2011 [10]. Furthermore, data covering the world’s best runners displayed a predominance of Japanese athletes in the 100-km races between 1999 and 2015 [11]. In agreement with these findings, a study performed with the purpose to investigate the participation in 100-km events found that most finishers (73.5%) were from Europe, especially from France (30.4%), but Japanese women and men were the fastest [10].

In the beginning of ultra-marathon running, mainly men were participating [8,12]. Notwithstanding the published studies, few of them were developed to understand female participation and performance [13,14]. Though in the beginning of ultra-marathon running, virtually no women were participating, the percentage of female participation increased through the years to around 20% [13,14].

In this context, Krouse et al. investigated the motivational aspects related to women engagement in ultra-marathons, and they found that female athletes were task oriented, internally motivated, health and financially conscious individuals [15]. In addition, Knechtle et al. analyzed the participation and performance trends in 100-km ultra-marathoners from both sexes [16]. Exploring data from 96,036 subjects (88,286 men and 7750 women) from 67 countries, who successfully finished the ‘100 km Lauf Biel’ in Switzerland between 1956 and 2019, they reported that an increase in the number of participants in age groups 40–49, 50–59, and 60–69 years was observed among women, whereas athletes in age groups 20–29 and 30–39 years reached the peak of participation in the late 1980s, followed by a decrease or stabilization, respectively [16]. Furthermore, Switzerland, Germany, and France were the countries with the highest numbers of participants throughout the history of the race [16]. When all 100-km ultra-marathon races held between 1960 and 2019 were considered, women were able to close the gap to men in the youngest (20–29 years) and in the oldest (>90 years) age groups for recreational athletes, and in athletes older than 70 years for the top three athletes [12].

The 6-h ultra-marathon is the shortest time-limited ultra-marathon race [17], but little has been investigated regarding this race format, where only the age of peak performance in the context of longer time-limited ultra-marathons was determined [18,19]. Only one study investigated participation and performance trends in time-limited ultra-endurance races, including multi-day events, in runners younger than 19 years of age [20]. We have, however, no knowledge where these runners come from as we have from other more popular ultra-marathons, such as the 100-km ultra-marathon races [21], the ‘Comrades’ [22], or the ‘Spartathlon’ [23].

Despite the relevance of the mentioned studies for ultra-marathon running, there is a lack in the knowledge of the participation and performance trends for female and male athletes in 6-h ultra-marathon events as the shortest time-limited ultra-marathon race format. Therefore, the aims of this study were: (i) to investigate the trends of participation by countries for both women and men; (ii) to investigate the age of peak performance according the running classification; and (iii) to investigate the differences in the performance between countries in the 6-h ultra-marathons held world-wide, considering all the athletes, the 10-best, and the 100-best. Based upon the above-mentioned findings, we hypothesized that (i) there would be an increase in the number of female participants and finishers during the years, and (ii) Japanese and Russian athletes would be the fastest 6-h ultra-marathoners. 

## 2. Materials and Methods

### 2.1. Ethical Approval 

The institutional review board of St Gallen, Switzerland, approved this study (EKSG 1 June 2010). Since the study involved the analysis of publicly available data, the requirement for informed consent was waived.

### 2.2. Data

This is an exploratory study, using information obtained from an official webpage. Data were collected from the results section of the website of DUV (Deutsche Ultramarathon Vereinigung) [24]. All information was related to the available results for female and male participants in ultra-marathons with 6-h in duration between 1982 and 2020, given that, before 1982, no 6-h ultra-marathon was officially held and registered. The available information included the athlete’s name, date of birth, sex, ranking, athlete’s average running speed and ranking for age, country of birth, and year of competition. The athlete’s age was computed taking into account the date of birth and the date of the competition. The athletes were analyzed regarding ranking position and age categories. For ranking, athletes were classified consonant with their position in the race classification (1st–3rd; 4th–10th; and higher than 10th position) based upon previous studies [25,26]. Regarding age, the athletes were categorized in four different age groups, <30 years; 31–40 years; 41–50 years; and >50 years.

### 2.3. Statistical Analysis

Descriptive information was expressed as mean (and standard deviation) or frequency (percentage) for continuous and categorical data, respectively. Data normality distribution was tested by the Kolmogorov–Smirnov test. In both sexes, the years of 1982–2020 were broken in three different moments, to identify differences regarding the number of participations, using the Kruskal–Wallis test. Differences in the age of peak performance was tested through two approaches, following the purpose of Ericsson [27]: (i) considering the existence of differences in age consonant with the race position (1st–3rd; 4th–10th; >10th), and (ii) considering differences in performance (running speed (km/h)) in different age categories (<30 years; 31–40 years; 41–50 years; >50 years). Categories were defined based upon previous studies considering differences in race position [25,26]. Given that age and running speed did not present a normality distribution, the non-parametric Kruskal–Wallis test, followed by the Bonferroni Correction for multiple tests, were used to verify differences between groups, for each considered variable. Statistical Package for Social Sciences version 26^®^ (SPSS, IBM Corp., Armonk, NY, USA), was used for statistical analysis. 

To observe differences in the performance by countries, we adopted three approaches: (i) considering all the countries, for both sexes; (ii) considering athletes classified among the 10 best in each country, in countries with at least 10 athletes in the ranking; and (iii) considering athletes classified among the 100 best, in countries with at least 10 athletes in the ranking. Based upon previous findings, countries with less than 10 athletes were excluded [28]. Countries were dummy coded, and the fastest countries were set as a reference for female and male runners. One multiple regression model was built, with all the countries included, aiming to estimate differences in the performance for each country against the reference country. Age, year of the event, and the interaction ‘age × event year’ were included, except age in the male analysis. Regression analysis was performed using SuperMix software, and the significance level was set at 0.05.

## 3. Results

For the present study, the sample included 23,203 female ultra-runners, aged 18–83 years, and 87,264 male ultra-runners, aged 18–85 years, who were officially finishers in a 6-h ultra-marathon held between 1982 and 2020.

### 3.1. Participation by Year and Country

The last 30 years indicated an increase in the number of ultra-marathoners in both sexes (Figure 1). When years were divided in three different moments, the last decade was the decade with the highest number of female participation (2011–2020 = 17,676), compared to the period 2001–2010 (4773) and until 2000 (754) (H (2) = 844.8; *p* < 0.001). In the last decade, 15.9% of the female runners were participants in race events held in 2019 (*n* = 2824), being the year with the highest number of participants in the last 30 years. The year 2020 was followed by a break in the participation (*n* = 913). The male participation has changed for 4578 athletes until 2000, for 61,326 during 2011–2020, with significant differences (H (2) = 55,683.4; *p*-adjusted < 0.001). The men-to-women ratio has decreased along the years, with the lowest values being observed in 1985 (2.00), 1986 (2.50), and 2020 (2.73). 

Figure 2 presents the distribution by country and by decades (until 1990, 1991–2000, 2001–2010, and 2011–2020) for women. An increase in the number of countries and athletes’ participation occurred during these years. In Figure 1, 54 female runners were distributed between six countries, and 63% of them were from the United States. During 1991–2000, the highest percentage of ultra-runners were from Germany (25.9%), followed by France (13.4%), Belgium (9.3%), and Australia (9.1%). For 2001–2010, Germany, France, USA, Austria, and Italy were responsible for more than half of the ranked athletes (60.4%). In the last decade, 4773 athletes finished a 6-h ultra-marathon, and the country with the highest number was Germany (16.9%), followed by the USA (16.1%), Italy (13.4%), and France (7.2%). Considering all the years (1982–2020), the highest percentage of the athletes was from Germany (18.8%), USA (14.6%), Italy (11.5%), and France (8.7%).

For men (Figure 3), until 1990, athletes were from 12 different countries, where 39.6% were from the USA. For the period 1991–1999, Belgium, France, Germany, and the Netherlands comprised more than 60% of the athletes. For the period 2000–2010, the athletes were from 55 different countries, with the highest numbers from Germany (20.3%), France (18.2%), and the Netherlands (9.1%). In the last nine years, athletes from the 98 different countries were competing, with the highest frequency of them from Italy (19.5%) and Germany (15.4%). Considering all the years (1982–2020), the highest percentage of the athletes came from Germany (16.6%), Italy (15.6%), France (12.5%), and the USA (8.0%).

### 3.2. The Age of Peak Performance

The mean age was 43.2 ± 9.3 years for female athletes (Figure 4a,b). When the sample was stratified consonant with running classification (1st–3rd; 4th–10th; and >10th position), differences according the age of peak performance were verified (H (2) = 416.6; *p*-adjusted < 0.001). The athletes classified in the 1st–3rd position were at a younger age (41.7 ± 8.9 years), compared to those classified in 4th–10th (43.6 ± 9.4 years; *p*-adjusted < 0.001), and higher than the 10th position (45.0 ± 9.7 years; *p*-adjusted < 0.001) groups. When athletes were stratified into different age categories, the age group 40–50 years was considered the reference. A non-significant difference in running speed was verified compared with athletes until 30 years (U = 43.4; *p*-adjusted = 1.000), but significant differences were observed when compared against the other age categories. 

For male athletes, the mean age was 46.09 ± 10.17 years (Figure 4c,d). Significant differences were found for age of peak performance consonant with running classification (H (2) = 3410.00; *p*-adjusted < 0.001). The athletes classified in the 1st–3rd position were at a younger age (41.5 ± 8.9 years), compared to the 4th–10th (44.2 ± 9.5 years; *p*-adjusted < 0.001), and higher than the 10th position (47.43 ± 10.24 years; *p*-adjusted < 0.001). When athletes were stratified in different age categories, the age group 40–50 years was considered as the reference. Significant statistically differences were observed in all age categories.

### 3.3. Performance by Country for Women

Descriptive statistics and regression results are presented in Table 1 and Appendix A. In total, the highest number of athletes were from Germany (18.8%), the USA (14.54%), Italy (11.48%), France (8.68%), and Sweden (6.44%). There was a high heterogeneity in age, since a high variability was observed between and within countries. For the regression analysis results, Russia was considered as the country of reference, since athletes from this country presented the highest running speed (10.01 ± 1.28 km/h) alongside the studied years. Significant differences were observed for several countries. Age, event year, and the interaction ‘age × event year’ were included in the model, but only the event year showed a significantly association with running speed, where athletes decreased running speed along the years (β = −0.020; *p* < 0.005).

### 3.4. Performance by Country for Men

Descriptive statistics and linear regression results are presented in Table 2 and Appendix A. The highest number of athletes were from Germany (16.6%), Italy (15.6%), France (12.5%), the USA (8.0%), and Sweden (4.5%). A high heterogeneity between and within countries is observed. The best performance was verified for athletes from Tunisia, and this country was considered as the reference for male performance (12.16 ± 1.46 km/h). Significant differences were observed for several countries. Year of event (β = −0.037; *p* < 0.001) and the interaction ‘year of event versus age’ (β = −1.43; *p* < 0.001) showed a significant association with running speed. Along the years, and with increasing of age, a decrease in running speed was observed. 

When the top 10 women per country were considered, the fastest women were from Slovenia, followed by Norway, Poland, Spain, Belgium, and Russia (Table 3 and Appendix A). When the top 100 women per country were considered, the fastest women were from Russia, followed by Belgium, Iceland, Ukraine, Denmark, and Poland (Table 4 and Appendix A). When the top ten men were analyzed, the fastest runners were from Tunisia, followed by runners from Belgium, Russia, Norway, and Slovenia (Table 5 and Appendix A). When the top 100 men were analyzed, the fastest runners were from Tunisia, Belgium, France, Lithuania, and Latvia (Table 6 and Appendix A).

## 4. Discussion

The purposes of this study were: (i) to analyze the trends of female and male participation by nationality; (ii) to investigate the age of peak performance according the running classification; and (iii) to investigate the differences in the performance between countries in a 6-h ultramarathon, considering all the athletes, the 10-best, and the 100-best. We hypothesized that there would be an increase in the number of participants and finishers during the years, and that Japanese and Russian athletes would be the fastest. The most important findings were: (i) an increase in female and male participation across years; (ii) most female and male participants were from Germany; (iii) mean age was 43.2 ± 9.3 years for female, and 46.09 ± 10.17 years for male runners; (iv) athletes in younger age groups were faster than athletes in older age groups; and (v) women from Russia and men from Tunisia were the fastest.

### 4.1. Increase in Female and Male Participation across Years

An early finding was that participation for both sexes increased across the years, while the men-to-women ratio decreased along the years. It happened despite the absolute number of male runners being higher than female ones. The increase observed in women participation is, in relative terms, higher than in men, which can be linked to changes in a plethora of domains, that favor women training and race event participation, such as cultural, economic, and intrapersonal changes [29].

Results indicated an increase in the number of female and male participants in 6-h ultra-marathons during the period 1982–2020. The last decade (2010–2020) showed the highest number of participants, especially in the year 2019, with 15% of the total of participants in the last decade. A break in 2020 was verified, most probably due to the COVID-19 pandemic, which led to the cancelation and suspension of several athletic events in 2020. Especially in ultra-running events, a previous study showed that the numbers of events and the ratio finishes/events significantly decreased during the COVID-19 pandemic [30].

The increase in the number of participants was expected, as reported in previous studies [12,13,23,31]. Krouse et al. suggested that the number of ultra-marathons has tripled in the last 30 years [15]. These rises in participants are partially explained by an increase in the number of young runners [32,33], female runners [13,14], and ultra-marathoners of very old ages [12]. In general, the majority of runners in ultra-marathon races are men, and women account for a small percentage of them. For example, in the ‘Badwater’ ultra-marathon, ~19% were women [34]. In the ‘Marathon des Sables’, an event that covers 250 km in the Sahara Desert, 13% of the participants were women [35]. However, the men-to-women ratio has decreased during recent years [13]. 

A study developed to understand the enablers and barriers for ultra-marathon participation in both sexes showed that enablers and barriers were similar for male and female runners [36]. The lack of the time was the main barrier to train for an ultra-marathon, given the needs for negotiating time for running with family and work commitments. However, it is possible that women experience more resistance to negotiating-efficacy men, reducing the commitment in training time.

### 4.2. Most Participants Originated from Germany

A second finding was that for both women and men, the highest numbers of participants were from Germany. When we consider the historic trends, most female and male runners were from the USA in the beginning. Over years, more athletes from European countries competed in these races, and the USA is the second-most represented country among women. However, among the top countries, only European countries were represented, led by Germany. Regarding each country’s participation, we showed that most of the female athletes were from Germany (18.8%), followed by the USA (14.6%), Italy (11.5%), and France (8.7%). Considering all male athletes, the highest percentage was from Germany (16.6%), followed by Italy (15.6%), France (12.5%), and the USA (8.0%).

The participation of US-American athletes can be associated with the high number of events hosted in North American countries (i.e., USA and Canada) [30], facilitating the access of these athletes, the opposite of what is observed by European ultra-runners, who have to travel longer to foreign countries than runners from the USA to compete [37]. These results were similar to those found by Gerosa et al. [38] for 161-km ultra-marathons (100 miles ultra-marathons), where athletes from Europe increased their participation from 1.6% (1998) to 14.5% (2011), while a decrease in the participation of USA athletes was verified in the same period (1998, 89.6%; 2011, 75.9%). Similar results were found by Cejka et al. [10], studying participants in 100-km ultra-marathons, where most of the athletes were from Europe, especially from France (30.4%). The authors also showed that athletes from Japan, Germany, Italy, Poland, and the USA increased their participation during the period 1998–2011. Data covering 85% of the ultra-running around the world showed that France, the USA, and South Africa have the highest proportions of the world’s ultra-runners [37]. 

### 4.3. A Higher Age in Male Compared to Female Runners

A third finding was that mean age was higher in men compared to women. Differences in the age of peak running performance do exist. A study analyzing the age of peak marathon performance in 1-year and 5-year age intervals of 451,637 runners (i.e., 168,702 women and 282,935 men), who finished the ‘New York City Marathon’ between 2006 and 2016, showed that women achieved their best marathon race time around five years earlier in life than men in both 1-year and 5-year age intervals [39].

Two main approaches were used to investigate the age of peak performance. The first approach, purposed by Lehmann [40], consisted of studying the age of the best athletes in the major competitive events around the world. The second approach consisted of analyzing the performances considering differences in the age groups [27]. We decided to use the two approaches to understand the age of peak performance. 

Firstly, the results showed differences in the age of peak performance when the sample was stratified by race position. Athletes classified in the 1st–3rd positions were younger than those in the 4th–10th and >10th positions. Generally, ultra-marathoners are aged around 40–50 years [18]. In the present study, the mean age was 43.2 ± 9.3 years for female and 46.09 ± 10.17 years for male runners. These results were according to previous research, where the age of peak ultra-marathon performance was ~42 years [6]. 

Secondly, given that most of the ultra-marathoners were aged between 40–50 years, it is possible that the results for the age of peak ultra-marathon performance are biased, especially when mean age is considered. In this sense, we performed an analysis to identify the difference in running speed for different age intervals. Athletes aged between 40 and 50 years was the reference, and the results showed non-statistically significant differences compared to those until 30 years. These results could be explained by the increase in the number of young female participants in the last decade [37]. Another point is the decrease in the physiological index associated with increasing age, which can impair performance in master athletes [41,42,43]. 

The performance of elite level master endurance athletes appeared to be maintained until ~35 years of age, followed by modest age-related decreases until ~50 years of age, with a progressive decrease in performance thereafter, with the greatest decline occurring after the age of 70 years in endurance running events [44,45]. In endurance running events, the decrease in performance is greater in women compared to men, possibly due to either biological or sociological differences [45]. Tanaka and Seals [45] suggested that these gender differences may partly be explained by a small number of female runners in the oldest age groups.

Endurance performance is dependent on the athlete’s ability to sustain power and overcome resistance or drag during competition. In endurance disciplines, these factors are interrelated with the systems that supply oxygen and energy substrates to the working muscles. The current research suggests that age-related reductions in endurance performance observed in master athletes are primarily related to age-related decreases in VO_2max_, followed by an age-related decrease in lactate threshold (LT) velocity, blood volume, and muscle mass. Pimental et al. [46] observed that VO_2max_ declined in endurance-trained athletes by 0.2 mL/kg/min per year between 20 and 50 years of age, increasing to a decline of 0.89 mL/kg/min per year between 50 and 74 years of age.

Moreover, the details about the contribution of central (i.e., cardiovascular) and peripheral (i.e., oxygen extraction) factors responsible for age-related decreases in VO_2max_ in master endurance athletes remains unclear. According to the Fick equation, maximal cardiac output, as well as maximal arterio-venous oxygen difference, have been found to decrease in master athletes [47]. The reduction in maximal cardiac output in master endurance athletes results from a reduction in both maximal heart rate and maximal stroke [48]. At a peripheral level, maximal arterio-venous oxygen difference has been found to modestly decline with age in master runners [49]. However, Tanaka and Seals [47] reported that maximal oxygen delivery is the major contributor to age-related decline in maximal arterio-venous oxygen difference in master endurance athletes.

Even though a high VO_2max_ be a prerequisite for success in endurance sports, the ability to sustain a high percent of VO_2max_ without accumulating fatigue is of greater importance. This is why LT can be a useful discriminator of endurance performance. Coyle [50] identified that LT is a major determining factor in endurance performance. Several studies [51,52] have consistently shown that both VO_2_ at LT and velocity at LT are better predictors of endurance running performance than VO_2max_ in younger distance runners. However, in older runners, it appears that endurance performance is correlated with both VO_2max_ and velocity at LT in male [53,54] and female runners [55,56].

Allen et al. [57] observed that male master runners had a significantly lower VO_2max_ (by 9%) than young runners, with no significant differences in running economy. Both groups attained a 2.5-mmol/L blood lactate level at the same running speed and VO_2_ during steady-state exercise. Similarly, Wiswell et al. [58] observed that LT as %VO_2max_ did not differ between male and female endurance runners, and increased significantly with age in both groups.

It is worth mentioning that the results of several studies [59,60,61] suggested that high-intensity training and maintenance of training volume may mediate the age-related declines in endurance performance and its physiological determinants. It can be concluded that a high intensity, as well as high volume, of training is important in maintaining or attenuating age-related decreases in endurance performance.

### 4.4. The Fastest Women Were from Russia and the Fastest Men from Tunisia

The analysis of the country with the fastest female runners led to different results when all, the top ten, or the top 100 women were investigated. When we considered all women and all countries, the fastest women were from Russia, followed by runners from Cyprus, Ukraine, Island, and Belgium. When the top 10 women per country were considered, the fastest women were from Slovenia, followed by runners from Norway, Poland, Spain, Belgium, and Russia. When the top 100 women per country were considered, the fastest women were from Russia, followed by Belgium, Iceland, Ukraine, Denmark, and Poland. Overall, it seems that women from Russia were the fastest. Considering all women, results showed that, from some countries, only very few athletes participated. For example, Cyprus, with 2 athletes, was second place in the ranking of the fastest athletes, behind Russia, with 452 runners. Cyprus was no longer in the analysis when the fastest 10 and the fastest 100 female runners were considered.

Also, the analysis of the country with the fastest male runners led to different results when all, the top ten, or the top 100 men were investigated. When we considered all men and all countries, the fastest men were from Tunisia, followed by athletes from Sri Lanka, Malta, Cape Verde, and Montenegro. When the top ten men were analyzed, the fastest runners were from Tunisia, followed by runners from Belgium, Russia, Norway, and Slovenia. When the top 100 men were analyzed, the fastest runners were from Tunisia, Belgium, France, Lithuania, and Latvia. Overall, it seemed that men from Tunisia were the fastest. Results considering all men showed that, from some countries, only very few athletes participated. For example, Sri Lanka, Malta, Cape Verde, and Montenegro, with less than 10 athletes, dropped out in the analysis when the top 10 and top 100 runners by country were considered.

Results of the linear regression indicated that athletes impaired their performance along the years. These results were similar compared to an analysis with 5,010,730 results from 15,451 ultra-running events reporting that athletes had added 1:41 min/mile to their average pace, which is a slowdown of 15% since 1996 [37]. 

Between-countries differences showed that Russian women were those with the best performance (10.01 ± 1.28 km/h) in the 6-h events during the last 30 years. These results partially agree with the author’s hypothesis, given that Russian athletes were the best in ultra-marathon running [62]. Similar results were found by Nikolaidis et al. [11] in an analysis involving athletes during 1999–2015, where the best 100-km athletes classified by IAAF (World Athletics) were from the Russia and Japan. Results reported by Cejka et al. [10] indicated that Japanese were the fastest athletes in 100-km ultra-running. Further, in a report realized in association with the International Association of Ultra-runners, it was verified that, in distance races longer than 26.2 miles (i.e., trail runs, mountain runs, and road runs), the fastest female athletes were from South Africa (~11:11 min/mile), Sweden (~12:27 min/mile), and Germany (~12:33 min/mile) [37].

The finding that men from Tunisia were the fastest in 6-h ultra-marathons was not expected, since we hypothesized that both female and male Russian athletes would be the fastest. Up to date, Tunisian athletes were only very marginally mentioned in long-distance swimming [63] and in marathon running [64]. No scientific studies are available on medium- and long-distance Tunisian runners. Our finding could be linked to the interference of physiological, environmental, social, and economic factors. The rural environment is predominant in the north, center, and south of Tunisia, with mountains, forests, and deserts, which is why there are a lot of transportation and infrastructure problems [65]. As such, a higher proportion of Tunisians cover greater distances to fulfill their needs. More particularly, most of the medium- and long-distance male Tunisian athletes came from rural areas. A higher proportion of them walked and ran longer distances to get to school each day. Although no scientific data are available, running in the south of Tunisia is popular as a school activity (i.e., due to the lack of materials to play other sports), and is also common due to the lack of infrastructure for transport (i.e., the distance is about 4 to 10 km from home to school) [66]. 

Also, in recent years, an increased number of participants in running activities was observed in Tunisia, and some international running events (e.g., ‘Sfax Marathon Olive Trees’) were organized, which could have increased the popularity of this activity [67]. In Tunisia, only a few half-marathons, marathons, and ultra-marathons are held [68]. Regarding ultra-marathons, the ‘Ultra Mirage^®^ El Djerid 100 km’ is the first 100km Ultra Trail taking place in the stunning Tunisian Sahara Desert [69]. The ‘100 km del Sahara’ is a further ultra-marathon held in Tunisia [70]. So, future studies are necessary to better explain why Tunisian men are the fastest in 6-h ultra-marathons, and whether they also perform the best in other distance-limited ultra-marathon races, such as 12-h and 24-h.

Knowledge about factors predisposing to achieve the best results in ultra-marathon running, although increasing, is still not definitively established [71]. According to Gajda et al., the mosaic theory of success in ultramarathon running (multifactorial, complex nature of the cause of achieved results or successes) additionally includes, among others, genetic factors found extensively in some populations, such as the presence of mtDNA haplogroup H (HV0a1 subgroup, belonging to the HV cluster), a characteristic of athletes with the highest endurance [72]; and normal pain resistance [73], not studied among the athletes described in this article. However, none of these factors guarantee success for an individual athlete or a given community (nation) in ultra-marathons.

### 4.5. Limitations

This study was not free of limitations. Firstly, the authors could not control the differences in altimetry and environmental conditions that can change during the years and impair the performances. Secondly, the database did not include information about the place of the ultra-running events, and these results can be associated with a large number of participants, due to the access and facilities to participation. Thirdly, the data did not provide information about the training characteristics and running experience, which can have an influence on the performance results. On the other hand, few studies have been developed to understand the participation and performance of women in ultra-running events. To the best of the authors’ knowledge, this is the first study to investigate the trend in participation, the age of peak performance, and the performance itself in 6-h female ultra-runners from the last 30 years. The results section of the website of DUV (Deutsche Ultramarathon Vereinigung) [24] might not be complete to consider all races worldwide.

## 5. Conclusions

The last forty years showed an increase in the number of 6-h ultra-marathons, with a higher increase in the number of female runners compared to male. Most of the participants were from Germany, the fastest women were from Russia, and the fastest men were from Tunisia. The mean age was higher in men compared to women. Future studies need to investigate whether Tunisian men were also the best in other distance-limited ultra-marathon races, such as 12-h and 24-h.

## Figures and Tables

**Figure 1 medicina-58-00179-f001:**
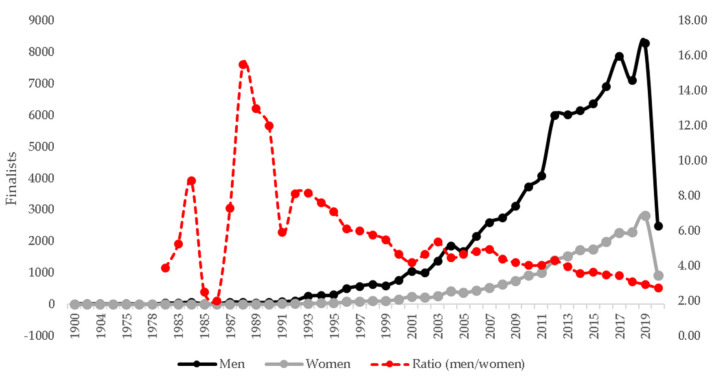
Sex participation and ratio men/woman during the 6-h ultra-marathons across the world.

**Figure 2 medicina-58-00179-f002:**
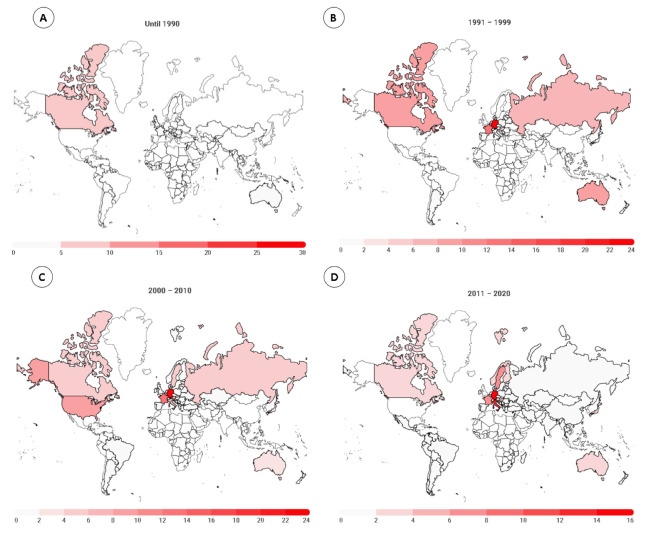
Female runners’ distribution by countries in 6-h ultra-marathons, by year groups ( (**A**) until 1990; (**B**) 1991–2000; (**C**) 2001–2010; and (**D**) 2011–2020).

**Figure 3 medicina-58-00179-f003:**
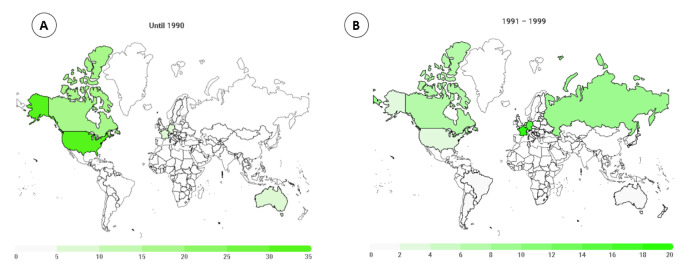
Male runners’ distribution by countries in 6-h ultra-marathons, by year groups ( (**A**) until 1990; (**B**) 1991–1999; (**C**) 2000–2010; and (**D**) 2011–2020).

**Figure 4 medicina-58-00179-f004:**
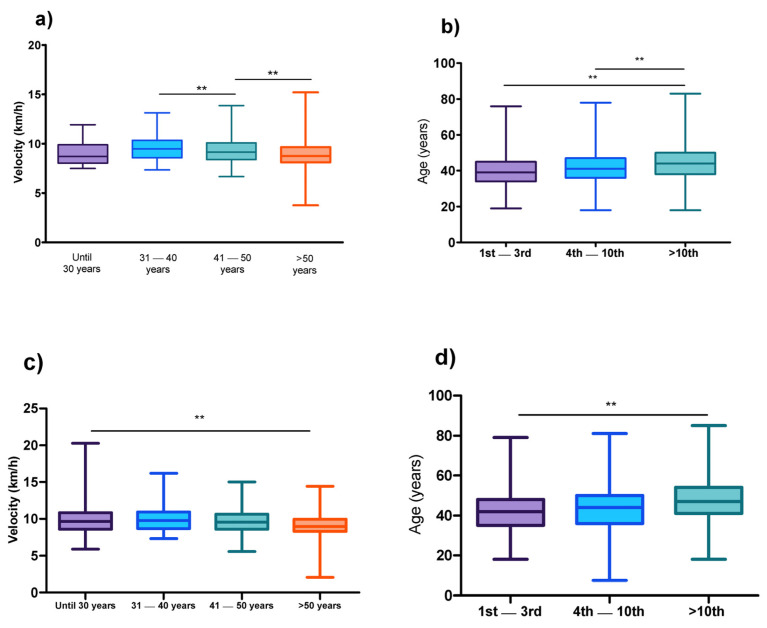
Age of peak performance. (**a**) Athlete’s performance in running speed, considering age interval—women; (**b**) differences in age of peak considering the athletes race position—women; (**c**) athlete’s performance in running speed, considering age interval—men; (**d**) differences in age of peak considering the athletes race position—men. ** *p*-adjusted below than 0.001.

**Table 1 medicina-58-00179-t001:** Descriptive statistics (mean and standard deviation) and linear regression results to verify the predictors associated with running speed in female ultra-marathoners from different countries (the top-10 countries, based on mean runner’s performance).

	β	*p*-Value	Frequency (%)	Age (Years)	Distance (km)	Running Speed (km/h)
				Mean (Std)	Mean (Std)	Mean (Std)
Russia	Reference	Reference	452 (1.95%)	38.89 (6.19)	60.06 (7.71)	10.01 (1.28)
Cyprus	−0.05736	0.93958	2 (0.01%)	34 (2.83)	57.11 (6.13)	9.93 (0.68)
Ukraine	−0.39729	0.02272 *	42 (0.18%)	40.32 (11.16)	56.71 (7.66)	9.86 (2.14)
Island	−0.32848	0.31361	11 (0.05%)	33.18 (5.53)	58.35 (3.86)	9.72 (0.64)
Belgium	−0.1956	0.01131 *	339 (1.46%)	44.58 (7.5)	58.2 (7.45)	9.7 (1.24)
Monaco	−0.05927	0.95578	1 (0%)	41	58.15	9.69
Albania	−0.09429	0.81674	7 (0.03%)	43.86 (1.07)	57.78 (6.17)	9.63 (1.03)
Denmark	−0.1359	0.05213	615 (2.65%)	43.11 (8.17)	53.69 (6.31)	9.58 (1.16)
Poland	−0.16411	0.20192	98 (0.42%)	38.15 (7.71)	57.27 (8.43)	9.55 (1.4)
Croatia	−0.24846	0.16918	43 (0.19%)	35.13 (7.78)	50.11 (4.49)	9.52 (1.02)
Constant	52.8911	<0.001 *	-	-	-	-
Age	0.10452	0.71356	-	-	-	-
Year event	−0.02093	0.00071 *	-	-	-	-
Year × Age	−0.00006	0.66075	-	-	-	-

Note: * indicates statistically significant differences for running speed, compared to Russia. Results are presented as mean and standard deviation.

**Table 2 medicina-58-00179-t002:** Descriptive statistics (mean and standard deviation) and linear regression results to verify the predictors associated with running speed in male ultra-marathoners from different countries (the top-10 countries, based on mean runner’s performance).

	β	*p*-Value	Frequency (%)	Age (Years)	Distance (km)	Running Speed (km/h)
				Mean (Std)	Mean (Std)	Mean (Std)
Tunisia	Reference	Reference	17 (0.0%)	44.82 (4.59)	72.95 (8.02)	12.16 (1.46)
Sri Lanka	−0.023	0.986	1 (0.0%)	36.00	71.42 (3.95)	11.90 (11.90)
Malta	0.177	0.804	4 (0.0%)	61.00 (0.82)	69.19 (10.39)	11.53 (0.75)
Cape Verde	−0.708	0.181	9 (0.0%)	35.22 (4.29)	67.82 (7.71)	11.30 (1.46)
Montenegro	−1.127	0.240	2 (0.0%)	40.50 (7.78)	65.70 (8.31)	10.95 (2.29)
Madagascar	−1.452	0.006 *	11 (0.0%)	37.44 (7.00)	65.01 (8.40)	10.83 (1.64)
Macau	−1.169	0.376	1 (0.0%)	-	64.88	10.81
Belgium	−1.374	<0.001 *	2784 (3.2%)	45.61 (9.59)	64.13 (9.84)	10.69 (1.64)
France	−2.225	<0.001 *	10932 (12.5%)	46.68 (9.64)	57.79 (9.34)	9.63 (1.32)
Faroe Islands	−0.919	0.159	5 (0.0%)	52.80 (2.59)	63.63 (8.85)	10.61 (1.37)
Constant	87.607	<0.001	-	-	-	-
Event year	−0.037	<0.001 *	-	-	-	-
Age × year	−1.43	<0.001 *	-	-	-	-

Note: * indicates statistically significant differences for running speed, compared to Tunisia. Results are presented as mean and standard deviation.

**Table 3 medicina-58-00179-t003:** Descriptive statistics (mean and standard deviation) and linear regression results to verify the predictors associated with running speed in the top 10 female ultra-marathoners from different countries.

	β	*p*-Value	Age (Years)	Distance (km)	Running Speed (km/h)
			Mean (Std)	Mean (Std)	Mean (Std)
Slovenia	Reference	Reference	39.06 (6.06)	68.28 (6.33)	11.38 (1.06)
Norway	0.25	<0.001 *	41.23 (10.25)	66.24 (6.17)	11.04 (1.03)
Poland	0.54	<0.001 *	37.83 (7.25)	65.55 (9.16)	10.92 (1.53)
Spain	−0.06	0.62	38.15 (8.08)	64.90 (5.97)	10.82 (0.99)
Belgium	0.57	<0.001 *	42.02 (5.99)	64.55 (6.93)	10.76 (1.16)
Russia	0.80	<0.001 *	36.57 (10.84)	63.75 (8.41)	10.62 (1.40)
Sweden	0.09	0.10	37.67 (8.57)	63.63 (6.59)	10.61 (1.10)
Hungry	0.21	<0.001 *	38.27 (6.41)	63.52 (6.45)	10.59 (1.08)
Netherlands	0.28	<0.001 *	45.36 (7.23)	63.29 (6.39)	10.55 (1.06)
Austria	0.19	<0.001 *	41.35 (8.69)	62.74 (6.74)	10.46 (1.12)
Constant	70.24	<0.001 *	-	-	-
Event year	−0.03	<0.001 *	-	-	-
Age × Year	0.00	<0.001 *	-	-	-

Note: * indicates statistically significant differences for running speed, compared to Slovenia. Results are presented as mean and standard deviation. Results show the first 10 countries (see Appendix A).

**Table 4 medicina-58-00179-t004:** Descriptive statistics (mean and standard deviation) and linear regression results to verify the predictors associated with running speed in the top 100 female ultra-marathoners from different countries.

	β	*p*-Value	Age (Years)	Distance (km)	Running Speed (km/h)
			Mean (Std)	Mean (Std)	Mean (Std)
Russia	Reference	Reference	42.27 (8.56)	53.82 (6.06)	8.97 (1.01)
Belgium	−0.034	0.644	42.30 (10.41)	54.08 (6.59)	9.01 (1.10)
Iceland	−0.171	0.598	43.19 (8.77)	56.04 (6.84)	9.34 (1.14)
Ukraine	−0.235	0.175	44.45 (7.45)	58.52 (7.50)	9.75 (1.25)
Denmark	0.026	0.694	41.37 (9.66)	52.58 (5.46)	8.76 (0.91)
Poland	−0.003	0.984	39.03 (7.89)	53.28 (5.41)	8.88 (0.90)
Netherland	−0.219	0.001	42.03 (9.51)	53.58 (6.15)	8.93 (1.02)
Croatia	−0.089	0.620	38.59 (8.20)	54.85 (6.30)	9.14 (1.05)
Romania	−0.237	0.116	34.25 (8.00)	57.31 (6.62)	9.55 (1.10)
Norway	−0.197	0.005 *	35.27 (8.82)	53.68 (6.45)	8.95 (1.07)
Constant	58.520	<0.001 *	-	-	-
Event year	−0.024	<0.001 *	-	-	-
Age × Year	−1.051	<0.001 *	-	-	-

Note: * indicates statistically significant differences for running speed compared to Russia. Results are presented as mean and standard deviation. Results show the first 10 countries (see Appendix A).

**Table 5 medicina-58-00179-t005:** Descriptive statistics (mean and standard deviation) and linear regression results to verify the predictors associated with running speed in the top 10 male ultra-marathoners from different countries.

	β	*p*-Value	Age (Years)	Distance (km)	Running Speed (km/h)
			Mean (Std)	Mean (Std)	Mean (Std)
Tunisia	Reference	Reference	45.47 (3.34)	75.52 (4.64)	12.59 (0.77)
Belgium	1.97	<0.001 *	43.59 (8.10)	73.50 (7.73)	12.25 (1.29)
Russia	1.54	<0.001 *	40.10 (11.08)	70.84 (9.06)	15.54 (11.81)
Norway	1.67	<0.001 *	43.01 (9.88)	70.11 (7.19)	14.80 (11.69)
Slovenia	1.38	<0.001 *	39.05 (8.73)	69.11 (7.17)	11.52 (1.19)
Lithuania	1.51	<0.001 *	40.59 (7.25)	68.71 (6.71)	14.88 (11.45)
Netherland	1.10	<0.001 *	46.24 (8.40)	62.35 (5.82)	14.54 (11.19)
Poland	1.20	<0.001 *	37.98 (9.12)	67.10 (10.09)	15.37 (11.18)
Spain	1.13	<0.001 *	42.23 (8.94)	66.80 (8.15)	11.13 (1.36)
Hungary	0.97	<0.001 *	39.91 (7.83)	66.26 (7.54)	11.04 (1.26)
Constant	86.62	<0.001 *	-	-	-
Event year	−0.04	<0.001 *	-	-	-
Age × Year	0.00	<0.001 *	-	-	-

Note: * indicates statistically significant differences for running speed compared to Tunisia. Results are presented as mean and standard deviation. Results show the first 10 countries (see Appendix A).

**Table 6 medicina-58-00179-t006:** Descriptive statistics (mean and standard deviation) and linear regression results to verify the predictors associated with running speed in the top 100 male ultra-marathoners from different countries.

	β	*p*-Value	Age (Years)	Distance (km)	Running Speed (km/h)
			Mean (Std)	Mean (Std)	Mean (Std)
Tunisia	Reference	Reference	44.82 (4.59)	72.95 (8.78)	12.16 (1.46)
Belgium	0.62	<0.001 *	45.60 (9.53)	64.78 (9.63)	10.80 (1.60)
France	−0.32	<0.001 *	46.68 (9.61)	57.85 (7.94)	9.64 (1.32)
Lithuania	0.67	<0.001 *	41.43 (7.68)	62.76 (8.36)	0.46 (1.39)
Latvia	0.43	0.03 *	42.86 (8.96)	61.99 (8.81)	10.33 (1.47)
Hungary	0.02	0.81	41.27 (8.57)	60.11 (8.19)	10.02 (1.36)
Iceland	−0.10	0.80	45.33 (7.81)	59.93 (5.28)	9.99 (0.88)
Denmark	−0.02	0.85	42.86 (7.91)	59.72 (7.33)	9.95 (1.22)
Bosnia	−0.15	0.52	40.19 (10.52)	59.18 (9.76)	9.86 (1.63)
Austria	−0.15	0.08	43.67 (9.76)	59.41 (8.25)	9.90 (1.37)
Constant	83.26	<0.001 *	-	-	-
Event year	−0.04	<0.001 *	-	-	-
Age × year	0.00	<0.001 *	-	-	-

Note: * indicates statistically significant differences for running speed, compared to Tunisia. Results are presented as mean and standard deviation. Results show the first 10 countries (see Appendix A).

## Data Availability

The athlete data was downloaded from the official website of DUV (Deutsche Ultramarathon Vereinigung) (https://statistik.d-u-v.org/geteventlist.php, accessed 26 December 2021).

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
