# Peer review of "The Sex Difference in 6-h Ultra-Marathon Running—The Worldwide Trends from 1982 to 2020"

_medicina, 2022, doi:10.3390/medicina58020179_

Round 1

Reviewer 1 Report

Reviewing 11 jan 2022. Journal Medicina (MDPI)

Title : The sex difference in 6-hours ultra-marathon running – the worldwide trends from 1982 to 2020

The purpose of this study was to investigate the trends in 6-hours ultra-marathon races from 1982 to 2020 for female and male ultra-runners, the participation and performance by countries, the age of peak performance and the differences in performance regarding countries.

Comments

  1. L35, L128 not trough but through
  2. L162 : authors explain in L162 that the men-to-woman ratio has increased along the years, with the highest values in 1986 (0.50), 1985 (0.40) and 2020 (0.37). Is there an explanation ?
  3. Figure1 shows Sex participation and verified through an exponential formula. For women I agree over three decades. For men the exponential formula should be applied until 2011. After there are high values (considered as outliers) disturbing this formula. Your exponential formula is not robust to outliers. The modeling shoud be different. Authors do not compare data (Sex participation) and exponential formula.
  4. L197 : not U but mu
  5. L227-228 : not B but beta

The remaining of the paper is relevant.

Author Response

Reviewer #1

Title: The sex difference in 6-hours ultra-marathon running – the worldwide trends from 1982 to 2020

The purpose of this study was to investigate the trends in 6-hours ultra-marathon races from 1982 to 2020 for female and male ultra-runners, the participation and performance by countries, the age of peak performance and the differences in performance regarding countries.

Comments

Reviewer comment: L35, L128 not trough but through

Author’s answer: Adjusted

Reviewer comment: L162: authors explain in L162 that the men-to-woman ratio has increased along the years, with the highest values in 1986 (0.50), 1985 (0.40) and 2020 (0.37). Is there an explanation?

Author’s answer: We adjusted the sentence and included the explanation in the discussion section.

Reviewer comment: Figure1 shows Sex participation and verified through an exponential formula. For women I agree over three decades. For men the exponential formula should be applied until 2011. After there are high values (considered as outliers) disturbing this formula. Your exponential formula is not robust to outliers. The modeling should be different. Authors do not compare data (Sex participation) and exponential formula.

Author’s answer: Thank for your comment. However, figure 1 just shows descriptive information, namely the absolute number of participants in each year (per sex), and the ratio men/women in each year.

Reviewer comment: L197: not U but mu

Author’s answer: Thank for your comment. However, we kept the use of “U”, since it is the notation used for the U-Man-Whitney test.

Reviewer comment: L227-228: not B but beta

Author’s answer: We included the symbol β

The remaining of the paper is relevant.

Reviewer 2 Report

Congrats to the authors of the idea. The paper is written in an interesting way. The methodology and statistical analysis used appears to be correct.   The work contains minor editorial errors:
- In all tables, please do not use bold to indicate significant results. Select them with e. g. "*" and add a legend.
- Please do not use bold in the main text next to "Figure 1", "Figure 2", etc. Correct throughout the text.
- Reference - #66 - has an incorrect entry. Please correct.
Thank you for the opportunity to conduct a review.

Author Response

Reviewer #2

Congrats to the authors of the idea. The paper is written in an interesting way. The methodology and statistical analysis used appears to be correct.   The work contains minor editorial errors:

Reviewer comment: In all tables, please do not use bold to indicate significant results. Select them with e. g. "*" and add a legend.

Authors answers: Thanks for your suggestion. We adjusted the tables accordingly.

Reviewer comment: Please do not use bold in the main text next to "Figure 1", "Figure 2", etc. Correct throughout the text.

Authors answers: Adjusted

Reviewer comment: Reference - #66 - has an incorrect entry. Please correct.

Authors answers: we corrected as suggested, this was a personal communication from one of the co-authors. We included one more reference in the manuscript, so the mentioned reference is ‘67’.

Reviewer comment: Thank you for the opportunity to conduct a review.

Authors answers: Thanks for your comments to improve the quality of the manuscript.